# The GABAergic System and Endocannabinoids in Epilepsy and Seizures: What Can We Expect from Plant Oils?

**DOI:** 10.3390/molecules27113595

**Published:** 2022-06-04

**Authors:** Fábio Rodrigues de Oliveira, Nágila Monteiro da Silva, Moisés Hamoy, Maria Elena Crespo-López, Irlon Maciel Ferreira, Edilene Oliveira da Silva, Barbarella de Matos Macchi, José Luiz Martins do Nascimento

**Affiliations:** 1Programa de Pós-Graduação em Neurociências e Biologia Celular, Instituto de Ciências Biológicas, Universidade Federal do Pará, Belém 66075-110, Brazil; oliveirafabio.fr@gmail.com (F.R.d.O.); nagilamonteiro.s@gmail.com (N.M.d.S.); edilene@ufpa.br (E.O.d.S.); 2Laboratório de Controle de Qualidade e Bromatologia, Curso de Farmácia, Departamento de Ciências Biológicas e da Saúde, Universidade Federal do Amapá, Macapá 68902-280, Brazil; 3Laboratório de Neuroquímica Molecular e Celular, Instituto de Ciências Biológicas, Universidade Federal do Pará, Belém 66075-110, Brazil; barbarella@ufpa.br; 4Laboratório de Farmacologia e Toxicologia de Produtos Naturais, Instituto de Ciências Biológicas, Universidade Federal do Pará, Belém 66075-110, Brazil; hamoy@ufpa.br; 5Programa de Pós-Graduação em Farmacologia e Bioquímica, Instituto de Ciências Biológicas, Universidade Federal do Pará, Belém 66075-110, Brazil; ecrespo@ufpa.br; 6Laboratório de Farmacologia Molecular, Instituto de Ciências Biológicas, Universidade Federal do Pará, Belém 66075-110, Brazil; 7Programa de Pós-Graduação em Ciências Farmacêuticas, Departamento de Ciências Biológicas e da Saúde, Universidade Federal do Amapá, Macapá 68902-280, Brazil; irlon.ferreira@gmail.com; 8Laboratório de Biocatálise e Síntese Orgânica Aplicada, Departamento de Ciências Exatas e Tecnológicas, Universidade Federal do Amapá, Macapá 68902-280, Brazil; 9Laboratorio de Biologia Estrutural, Instituto de Ciências Biológicas, Universidade Federal do Pará, Belém 66075-110, Brazil; 10Instituto Nacional de Ciência e Tecnologia de Biologia Estrutural e Bioimagem (INCT-INBEB), Rio de Janeiro 21941-590, Brazil; 11Instituto Nacional de Ciência e Tecnologia em Neuroimunomodulação (INCT-NIM), Rio de Janeiro 21040-900, Brazil

**Keywords:** GABAergic system, seizures, epilepsy, plant oils, endocannabinoids

## Abstract

Seizures and epilepsy are some of the most common serious neurological disorders, with approximately 80% of patients living in developing/underdeveloped countries. However, about one in three patients do not respond to currently available pharmacological treatments, indicating the need for research into new anticonvulsant drugs (ACDs). The GABAergic system is the main inhibitory system of the brain and has a central role in seizures and the screening of new ACD candidates. It has been demonstrated that the action of agents on endocannabinoid receptors modulates the balance between excitatory and inhibitory neurotransmitters; however, studies on the anticonvulsant properties of endocannabinoids from plant oils are relatively scarce. The Amazon region is an important source of plant oils that can be used for the synthesis of new fatty acid amides, which are compounds analogous to endocannabinoids. The synthesis of such compounds represents an important approach for the development of new anticonvulsant therapies.

## 1. Epilepsy and Seizures

Epilepsy is one of the most common serious neurological disorders, affecting approximately 50 million people worldwide, with an estimated 5 million new cases each year. Approximately 80% of these cases are in developing countries, probably due to poor medical infrastructure, increased risk of birth-related injuries (trauma, low weight, hypoxia,) and endemic diseases such as malaria and neurocysticercosis [1]. Epilepsy has many causes, including genetic and developmental abnormalities, brain injuries induced by ischemic stroke, intracerebral hemorrhage, infections, drug abuse, brain tumors, and a variety of psychological and social morbidities [2,3,4,5,6].

Epilepsy is defined as a chronic disorder characterized by abnormal brain electrical activity with spontaneous and recurrent seizures and often psychic manifestations, such as disturbances in cognition, behavior, consciousness, involuntary movements, and involvement of different brain areas [7,8]. One of the most common symptoms is seizures that directly influence patients’ quality of life and psychosocial functioning [9,10,11].

According to the International League Against Epilepsy (ILAE), seizures are divided into three categories: generalized, focal (previously called partial), and epileptic spasms [12,13].

Generalized seizures start in neuronal networks in both hemispheres and can be subdivided into absence seizures and generalized tonic-clonic (GTC), myoclonic and atonic seizures. Absence seizures are characterized by sudden lapses of consciousness, blinking eyes, or head movement, and their pathophysiology seems to be associated with rhythmic oscillations of the thalamus-cortical pathways with generalized spike and slow wave discharges and is generally associated with mutations of the Ca^2+^ channel [14,15]. GTC seizures consist of bilateral symmetrical convulsive movements of all limbs in addition to impairment of consciousness, unlike myoclonic seizures, which consist of sudden and involuntary movements that can affect one or several muscles, just as atonic seizures involve weakness of the muscle tone followed by drop attack [16]. Myoclonic and atonic seizures initially show a normal electroencephalogram followed by a generalized polyspike-and-wave epileptiform activity that precedes the onset of myoclonic-atonic and atonic seizures. The pathophysiology of these epileptic syndromes is more associated with voltage-gated Ca^2+^ channels and the GABAA receptor α1 subunit [15,17].

Focal seizures originating in neuronal networks are limited to a single cerebral hemisphere and can develop at any point in life. Focal epilepsies include unifocal and multifocal disorders, as well as crises involving a variety of types of epileptic seizures, and may include focal perceptual seizures, focal perceptive or impaired perception, focal motor seizures, and non-motor and focal seizures progressing sometimes to bilateral tonic-clonic seizures [18,19]. The interictal EEG typically shows focal epileptiform discharges and symptoms occur only when the seizure spreads to activate or disrupt cortical networks; as such, there is a wide spectrum of focal disease that depends not only on the location of the epileptic focus and its duration but also on the connected cortical and subcortical areas [18,19].

Epileptic spasms are a type of epileptic syndrome in children with infantile spasms and show changes in cerebral rhythms defined as hypsarrhythmia, with high-amplitude, arrhythmic, asynchronous delta activities, and multiple spikes. These epileptic spasms are associated with alterations in the gene regulation network of the GABAergic forebrain during development, and abnormalities in molecules expressed at the synapse [3,20,21].

## 2. Role of GABA in the Pathophysiology of Epilepsy and Seizures

GABA (γ-aminobutyric acid) and L-glutamate are the main inhibitory and excitatory neurotransmitters in the central nervous system (CNS), respectively, and both actively participate in epileptic disease [22,23]. However, the GABAergic system has been classically considered a main target of the anticonvulsant pharmacopeia, probably due to the high efficacy and potency of drugs acting on this system.

GABA-mediated inhibitory responses account for 30–40% of the synaptic connections in the brain [24] and play a fundamental role in the fine control of the excitability of circuits in the brain, including synchronization and generation of theta and gamma rhythms [23,25,26]. Modulation of GABA synaptic activity involves GABA synthesizing enzymes, transporters, GABA_A_, and GABA_B_ receptors generating diversity at GABAergic synapses. Alterations in the expression of these signaling components have been implicated in several disorders, including epilepsy [27,28].

GABA is synthesized from glutamate by the glutamic acid decarboxylase (GAD) enzyme [29]. GABA activity is rapidly terminated at the synapse by reuptake into nerve terminals and is metabolized by a reaction catalyzed by GABA transaminase (GABA-T) involved in the regulation of the GABAergic system [30,31]. GABA is released as a neurotransmitter into the synaptic cleft in many brain areas when stimulated by depolarization and exerts its effects pre- and postsynaptically via ionotropic (GABA_A_) and metabotropic (GABA_B_) receptors [22,24,32]. Ionotropic receptors (GABA_A_s) trigger the opening of chloride channels, resulting in membrane hyperpolarization of postsynaptic cells [24], while GABA_B_ receptors produce slow and prolonged inhibitory signals via G proteins and second messengers [33]. GABA transport is responsible for extracellular GABA clearance [34,35], a mechanism coupled to the cotransport of Na^+^ and Cl^−^ down their concentration gradient. The activities of these transporters determine the amount of available GABA for activation by presynaptic nerve terminals and glial cells. Disruption of these factors contributes to altered GABAergic transmission in epilepsy [22,26,36]. A large number of pro-convulsant drugs impair the inhibition mediated by GABA. Glutamic acid decarboxylase (GAD) is diminished in interneurons in discrete regions of the epileptogenic cortex and hippocampus, and high levels of the antibody-GAD are convulsants and produce cerebellar ataxia and temporal lobe epilepsy [26,36,37].

After GABA is released from vesicles into the synapse, its effects are modulated by ionotropic (GABA_A_) and metabotropic (GABA_B_) receptors. GABA_A_-Rs are comprised mainly of two α subunits, two β subunits, and one γ subunit. The heterogeneity of GABA_A_ receptors with their many allosteric sites is associated with the regulation of different GABAergic functions and they can be targeted by many drugs, such as benzodiazepines, barbiturates, neuroactive steroids, anesthetics, ethanol, and cannabinoids. The GABA_A_ receptor has a chloride ion channel that is gated by GABA and is blocked by convulsants [27,28,38]. 

GABA_B_ receptors are comprised of two GB1 and GB2 subunits. The GB2 subunit is responsible for the coupling of GABA_B_ receptors to G protein activation-associated Gi protein-activated K^+^ channels [39]. Many point mutations have been associated with genes encoding these receptor subunits that result in epileptic syndromes [38]. Effective GABA_B_ positive allosteric modulators are important for the treatment of seizure disorders. Another important target in the regulation of GABA activity is the GABA transporters that rapidly bind and remove GABA. Inhibition of GABA transporters by selective inhibitors such as tiagabine can be used as positive modulators that increase GABA extracellularly and are pharmacologically relevant to the control of epilepsy [35].

## 3. Treatment of Seizures and Screening of New Drugs

The treatment of epilepsies and seizures involves drugs in mono- or polytherapy, with first-generation anticonvulsant drugs (ACDs), such as phenytoin, barbiturate, and ethosuximide. Second-generation drugs, such as carbamazepine, valproic acid, and diazepam, constitute an important line of therapy and offer advantages over the former due to a lower risk of drug interactions and better tolerability. The third generation of ACDs still includes some drugs such as tiagabine, topiramate, pregabaline, and vigabatrine, which act on the GABAergic system as their main mechanism of action [40,41,42] (Figure 1). The continuing search for new anticonvulsant drugs is caused by the high prevalence of refractory epilepsy (when seizures are not controlled by the pharmacological treatment in use); about one in three patients does not respond to the available ACDs [41].

ACDs have diverse mechanisms of action, such as sodium or calcium channel blockers, agonists or potentiators of the effects on the GABAergic system, and blockers of the effects of glutamate via NMDA and AMPA receptors [43]. Unfortunately, some aspects that persist during treatment are a lack of substantial control of epileptic seizures [44].

In 60–80% of cases, seizures can be controlled with chronic use of these ACDs; however, all the ACDs currently available have adverse effects such as headache, fatigue, dizziness, drowsiness, and nausea; they can also cause neurological problems such as anxiety, depression, and sleep disorders, [45,46,47]. In addition, the chronic use of these drugs, especially GABAergic drugs, is limited due to frequent adverse effects, such as tolerance, dependence, and the presence of a withdrawal state, despite their high efficacy [48]. These numerous adverse effects are the main reason for the failure of compliance to the chronic treatment with ACDs in epilepsy, thereby explaining the need for the discovery of new anticonvulsant drugs with fewer adverse effects.

The study of new anticonvulsant drugs involves the use of several animal models that were developed with the aim of mimicking clinical symptoms and providing information about the mechanisms involved in the genesis and maintenance of seizures [49]. Diverse drugs such as pilocarpine or kainic acid have been used as chemoconvulsants in animal models of epilepsy for testing new ACD candidates [50]. One of the gold-standard models for the screening of new ACDs involves the chemical induction of seizures with pentylenetetrazol (PTZ), which is capable of generating acute seizures [51,52,53]. The mechanism of action of PTZ consists of noncompetitive blocking of GABA_A_ receptors through inhibition of the chloride channel associated with the receptor complex. Studies have shown that there is a specific binding of PTZ with the benzodiazepine site of the GABAergic receptor [9,51]. This inhibition prevents the action of this neurotransmitter and consequently its depressant effect on the CNS [54]. PTZ is the only chemoconvulsant that can be used for testing both isolated acute seizures (using a single administration of PTZ) and chronic and recurrent epilepsy (using the “kindling” model with repeated administration of low doses of PTZ). Seizures induced by PTZ start with myoclonic spasms, characterized by fleeting movements of muscle excitation or relaxation until a generalized tonic-clonic crisis occurs [55]. The PTZ model can be used to screen ACDs with different clinical effects and was crucial for the successful identification of methosuximide and ethosuximide, due to their superior tolerability [41]. Thus, this model is very useful for testing new drugs and/or bioproducts, including the endocannabinoids that have been already studied for their anticonvulsant action [9].

## 4. Endocannabinoids, Plant Oils, and Seizure Control

The endocannabinoid system (ECS) consists of endogenous cannabinoids, receptors, and metabolic enzymes. This system is present in most vertebrate species and is distributed in several organs and tissues, including the CNS [56]. Endocannabinoids are endogenous bioactive lipids that are produced locally through specific biosynthetic pathways. Endocannabinoids are divided into two classes, N-acylethanolamines (NAEs) and primary amides. Anandamide and 2-arachidonylglycerol (2-AG) (Figure 2), are lipid mediators derived mainly from arachidonic acid and are the most frequent biosynthetic routes [57,58,59]. Endocannabinoids, such as anandamide and its analogs, have been extensively studied as targets of new therapeutic options for disorders of the CNS, including the prevention of epileptic seizures [9,60].

Endocannabinoids are produced on demand from membrane phospholipids, with synthesis and degradation regulated by a set of enzymes that are widely distributed in the brain. Among these enzymes, diacylglycerol lipase (DAGL), monoacetylglycerol lipase (MAGL), and fatty amide hydrolase (FAAH) are of particular interest [61,62,63,64].

DAGL and phospholipase Cβ enzymes participate in the synthesis of 2-AG from membrane phospholipids. 2-AG is degraded mainly by the MAGL enzyme, producing arachidonic acid (AA) and glycerol. FAAH degrades anandamide to AA and ethanolamine [58]. These enzymes have crucial roles in the regulation of the tissue levels or actions of endocannabinoids.

Anandamide is produced in neurons during depolarization, and its biosynthesis involves a calcium-dependent transacylase (CDTA) that transfers the acyl group of phospholipids to the primary amine of phosphatidylethanolamine (PE) to generate n-acyl phosphatidylethanolamines (NAPEs), which are hydrolyzed by a type D phospholipase to generate NAEs [65]. This route may also involve other phospholipases (type C) with substrates such as n-arachidonoyl phosphatidylethanolamine [66].

2-AG is a monoacylglycerol molecule that acts as a retrograde messenger by activating both CB1 and CB2 cannabinoid receptors and is a key regulator of neurotransmitter release [58].

Endocannabinoids act on type 1 and type 2 receptors (CB1 and CB2) coupled to the Gi/o protein, and their binding leads to a decrease in the level of intracellular cyclic AMP and activation of mitogen-activated protein kinase, in addition to modulating potassium channels and inhibiting calcium channels [67,68,69]. These receptors are distributed in the CNS, mainly in the olfactory bulb, hippocampus, lateral striatum, and cerebellum, and in moderate amounts in areas of the cerebral cortex, hypothalamus, and spinal cord [25,70,71,72].

Among the cannabinoid receptors, CB1 has been identified as the main receptor responsible for the central effects of these groups of molecules [72,73,74]. Physiologically, endocannabinoids are produced and released by postsynaptic neurons and act on the presynaptic CB1 receptor, thus producing a stabilizing effect on the synapses, modulating the balance between excitatory and inhibitory neurotransmitters (glutamate and GABA, respectively) in the CNS [69,71,75], according to need, for example, in response to hyperexcitation [76,77,78].

CB2 receptors are seven transmembrane G protein-coupled receptors that are predominantly expressed in cells and tissues of the immune system [79]. However, CB2 has also been identified in areas of the CNS, mainly brainstem neurons, astrocytes, and microglial cells [80]. The CB2 receptor is overexpressed in response to diverse CNS insult and may play an important role in epilepsy [81].

## 5. Endocannabinoids from Plant Oils

Plant oils, frequently found in nature, are rich in free fatty acids with different applications. Triglycerides from plant oils, such as andiroba oil, can be used to produce fatty acid amides. Andiroba oil is extracted from the seeds of the andiroba tree (*Carapa guianensis Aublet*). Andiroba oil is comprised of triglycerides and fatty acids, predominantly oleic, palmitic, stearic, myristic, linoleic, and linolenic acids (Table 1) [82,83].

Fatty acid amides derived from these fatty acids are analogs of cannabinoids and have been widely studied as a new functional class of endogenous signaling molecules active in the endocannabinoid system [88,89].

Synthetic endocannabinoids, such as fatty acid amides, can be obtained from triglycerides and/or fatty acids in the presence of metallic catalysis [90,91] or biocatalysis [92,93]. Another possibility is based on the aminolysis of fatty acids (Figure 3), and their different pharmacological properties are attributed to the presence of the amide functional group in their molecules [94,95,96]. Aminolysis is carried out by an enzymatic reaction with excellent selectivity and conversion rates, mainly of amino alcohol compounds [97].

The fatty acids of Patawa oil (*Oenocarpus bataua*), have also been submitted to the amidation process, with the major products being the endocannabinoid analogs N-isopropylpalmitamide and N-isopropyloleamide [95]. Oils extracted from *Bertholletia excelsa* (Brazil nut) have also been a substrate for the production of endocannabinoid analogs using lipase from *Pseudomonas fluorescens*, with a yield of up to 95% after reaction [98].

## 6. Endocannabinoids and Seizure Control

During the onset of a seizure, 2-AG signaling is crucial for its suppression, which is related to the decrease in neuronal excitability due to its action on presynaptic CB1 receptors [72,99,100,101,102]. For this reason, natural or synthetic cannabinoids, as well as their analogs, have been explored due to their biological actions, such as anti-inflammatory, antioxidant, and neuroprotection, with recognized anticonvulsant properties [61,102,103,104,105,106,107,108,109,110].

Endocannabinoid analogs, such as palmitoylethanolamide and oleamide, have anticonvulsant properties and are capable of modulating the seizure threshold and decreasing neuronal excitability without any toxic effects [111,112,113,114].

The anticonvulsant activity of oleamide has already been tested in several induction models (picrotoxin, caffeine, strychnine, and PTZ) in Swiss mice at doses between 43.7 and 700 mg/kg. Promising results were found only in the PTZ model, with a dose-dependent response, and from 350 mg/kg, presenting results similar to diazepam (5 mg/kg), used as a control drug [115]. The site of action of oleamide is related to GABAA receptors at the binding site of benzodiazepines and displays GABA-induced Cl^−^ currents that are potentiated when administered together with GABA [116].

Another aspect that can be addressed in the relationship between endocannabinoids and anticonvulsant activity involves the enzymes that participate in their metabolism. Studies of the brains of epileptic patients, although they have shown no difference in the expression of the enzymes NAPE-PLD, MAGL, and FAAH relative to normal individuals, showed that the production of 2-AG may be compromised due to the lower expression of DAGL mRNA [117]. A previous study using knockout animals for this enzyme found that they presented with much more intense seizures, as well as higher levels of mortality when induced with kainite [101].

Likewise, in in vivo models using Wistar rats, the effects of enzymatic blockers on metabolism were evaluated in the PTZ model (85 mg/kg); the study showed that the blockade of the MAGL and ABH6 enzymes (responsible for the degradation of endocannabinoids) has an anticonvulsant effect [118]. It has also been shown that blocking FAAH activity indicates increased endocannabinoid activity, reducing seizures and the brain damage induced in an excitotoxicity model induced by kainate [119].

## 7. Conclusions

Thus, endocannabinoid fatty acid amides and their analogs, as well as other endocannabinoids, participate in the regulation of excitability in the synaptic cleft. Furthermore, plant oils, as sources of endocannabinoids, are natural bioactive products, and synthetic biochemistry may be a new approach for the development of therapeutic strategies for epilepsy, especially in low-income countries.

## Figures and Tables

**Figure 1 molecules-27-03595-f001:**
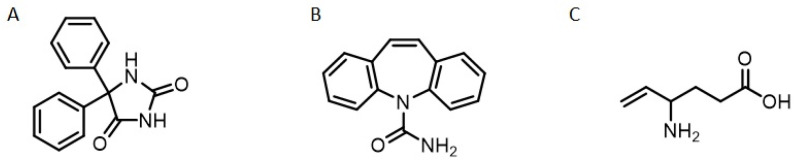
Chemical structure of clinical used anticonvulsant drugs in clinical use. (**A**) Phenytoin, first generation; (**B**) Carbamazepine, second generation, and (**C**) Vigabatrine, third generation.

**Figure 2 molecules-27-03595-f002:**
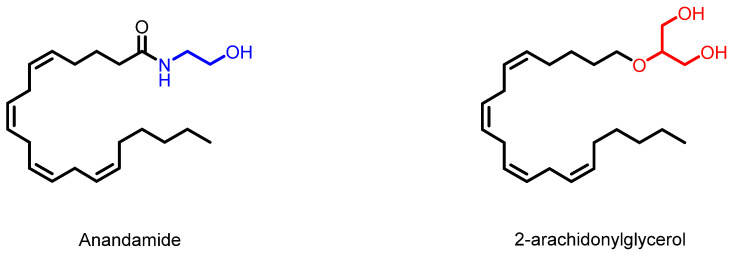
Examples of natural endocannabinoids.

**Figure 3 molecules-27-03595-f003:**
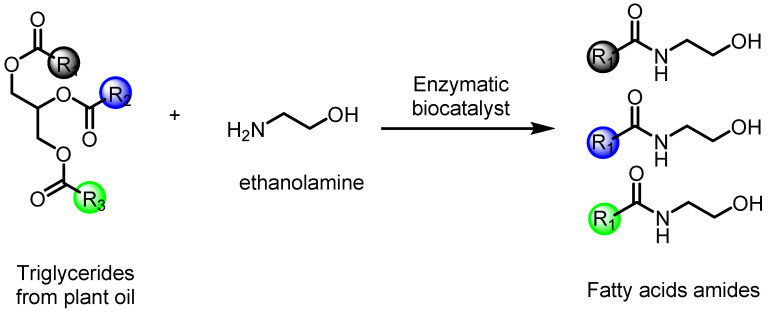
Synthesis of fatty acid amides by the reaction of aminolysis with triglycerides from plant oil and ethanolamine. R1, R2, and R3 are alkyl chains.

**Table 1 molecules-27-03595-t001:** Main fatty acids (%) in andiroba oil.

Fatty Acid and Structure						
Myristic C14:0	Palmitic C16:0	Stearic C18:0	Oleic C18:1	Linoleic C18:2	Linolenic C18:3	References
0.36	24.72	9.57	50.12	10.93	1.05	Pantoja et al., 2013 [84]
0.05	27.71	9.34	49.90	9.58	1.43	Araujo-Lima et al., 2018 [85]
-	31.02	10.53	42.71	12.93	tr.	Silva, 2018 [86]
-	27.30	12.52	47.19	9.29	-	Sousa et al., 2021 [87]

- means not detected and tr. means trace concentrations, under the limit of quantification.

## Data Availability

Not applicable.

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
