# Peer review of "The GABAergic System and Endocannabinoids in Epilepsy and Seizures: What Can We Expect from Plant Oils?"

_molecules, 2022, doi:10.3390/molecules27113595_

Round 1

Reviewer 1 Report

The manuscript is well organized, very clear and highlights the role of endocannabinoids in epilepsy.
Could the authors also add examples of other drugs used in epilepsy?
perhaps a graphical abstract would be appropriate, in order to have immediately the role of endocannabinoids

Author Response

Reply: we thank the reviewer with gratitude and agreement. We have added examples of other drugs used in epilepsy and also the graphic abstract as suggested by the reviewer. The drugs were added in the text (ethosuximide - first generation; diazepam - second-generation; topiramate and pregabaline - third-generation)

Reviewer 2 Report

Abstract: line 40, specify the importance of amazon (in other words, specify  amazon connection to plant oils and endocannabinoids) 

Page 2: line 57 to 62, revise the order of discussion as mentioned in the order in line 58. Also, mention the mechanism of action for the epileptic spasms like the other two types. 

line 61, modify the sentence: "spheres, and epileptic ............. spasms" to "spheres, and epileptic spasms, a type of epileptic seizures in children, and are usually accompanied by muscle spasms."

Page 2, line 63 to 76, there was a discussion only on generalized seizures and their subtypes. However, there was no discussion given about the other two types of seizures (focal and epileptic spasms)

page 2, line 90-91 the sentence has redundancy.

page 2, clarify line 100 in terms of grammar

page 3, line 111 modify the sentence "after GABA release......." to "after GABA is released ........ "

page 3, change the title of the section  "Treatment seizures and screening of new drugs" to "Treatment of seizures and screening of new drugs"

page 3, in the section of "Treatment seizures and screening of new drugs" better include the structures of the drugs (use chemdraw to draw the structures, not the images)

Page 4, 1st paragraph, there was a discussion about the seizure induction by PTZ for drug screening. However, the discussion about the drug screening is missing. Elaborate it.

Page 4, section 4, define endocannabinoids and endocannabinoid system in detail, also classify them here instead of explaining them in the section 5, page number 217 to 220.

Page 4, line 180-181, clarify why the enzymes are highlighted. Also explain in details about these enzymes.

Page 4, section 4, there was a discussion about the "Anandamide" however there is nothing discussed about 2-AG, which is focused in the section 6. I suggest the authors to discuss about 2-AG in detail next to the 3rd paragraph (after the discussion about anandamide)

page 4, section 4, there was a discussion on CB1, however, nothing was discussed about CB2, elaborate on it further.

Section 5 has to be rewritten focusing more on the endocannabinoids from plant oils. Most of the discussion is off topic and irrelevant. line 215 to 220 is more related to types of endocannabinoids, move this to section 4 at appropriate position. 

section 5, line 221 to 226, there was discussion focusing on synthesis of synthetic endocannabinoids, which is inappropriate in the context.

Also, line 230 to 232, there was discussion about the authors work, which is not relevant to the discussion (better have a separate section about the synthetic cannabinoids and add more information, not just authors own work.

Figures 1 and 2 use chemdraw to draw the chemical structures (not the pictures)

Also the entire paper should be thoroughly checked for the typos and grammatical errors.

Author Response

We thank the reviewer comments towards our work and the opportunity to revise it and to make it clear.

Also the entire paper should be thoroughly checked for the typos and grammatical errors.

Reply:  The manuscript has been carefully revised by a native English speaker to improve the grammar and readability. 

  1. Abstract: line 40, specify the importance of amazon (in other words, specify  amazon connection to plant oils and endocannabinoids) 

Reply: We replaced a sentence in the Abstract as shown to strengthen the connection of Amazon and plant oils and endocannabinoids.

Sentence removed: Although scientific literature on the anticonvulsant properties of endocannabinoids from plant oils are relatively scarce, it can be an important approach for the developing of new anticonvulsant therapies which also are accessible in low- and middle-income countries such as those in the Amazon.

Added sentence: however, studies on the anticonvulsant properties of endocannabinoids from plant oils are relatively scarce. The Amazon region is an important source of plant oils that can be used for the synthesis of new fatty acid amides, which are compounds that are analogous to endocannabinoids. The synthesis of such compounds represents an important approach for the development of new anticonvulsant therapies

  1. Page 2: line 57 to 62, revise the order of discussion as mentioned in the order in line 58. Also, mention the mechanism of action for the epileptic spasms like the other two types. 

Reply: We made the suggested alteration and added the sentence, as follows:

“…Epileptic spasms are a type of epileptic syndrome in children and related specifically to alterations in the gene regulation network of the GABAergic forebrain during development, and abnormalities in molecules expressed at the synapse.”

  1. line 61, modify the sentence: "spheres, and epileptic ............. spasms" to "spheres, and epileptic spasms, a type of epileptic seizures in children, and are usually accompanied by muscle spasms."

Reply: The sentence was modified.

  1. Page 2, line 63 to 76, there was a discussion only on generalized seizures and their subtypes. However, there was no discussion given about the other two types of seizures (focal and epileptic spasms)

Reply: We made the suggested alteration.:

  1. page 2, line 90-91 the sentence has redundancy.

Reply: We agree to the reviewer and removed part of the sentence to make the text concise and understandable.

  1. page 2, clarify line 100 in terms of grammar

Reply: We entirely agree with the reviewer and we decided to remove this statement to avoid confusion. 

  1. page 3, line 111 modify the sentence "after GABA release......." to "after GABA is released ........ "

Reply: Done.

  1. page 3, change the title of the section  "Treatment seizures and screening of new drugs"to"Treatment of seizures and screening of new drugs"

Reply: Done. 

  1. page 3, in the section of "Treatment seizures and screening of new drugs" better include the structures of the drugs (use chemdraw to draw the structures, not the images)

Reply: We thank the reviewer suggestion and now we provide the figure, now Figure 1, with a drug of each generation.

  1. Page 4, 1st paragraph, there was a discussion about the seizure induction by PTZ for drug screening. However, the discussion about the drug screening is missing. Elaborate it.

Reply: We thank the reviewer suggestion and added the sentence, as follows:

The PTZ model can be used to screen ACDs with different clinical effects and was crucial for the successful identification of methosuximide and ethosuximide, due to their superior tolerability [39].

  1. Page 4, section 4, define endocannabinoids and endocannabinoid system in detail, also classify them here instead of explaining them in the section 5, page number 217 to 220.

Reply: We thank the reviewer to call attention for that. The definition was added and the endocannabinoids classification was removed from section 5, as follow. We also inserted a figure with the natural cannabinoids (Figure 2).

Old sentence: Endocannabinoid system (ECS) is present in most vertebrate species and is distributed in several organs and tissues, including the CNS [52]. Endocannabinoids are endogenous lipid, such as arachidonoylethanolamide (anandamide - AEA) and 2-arachidonoyl-glycerol (2-AG), and their analogs are the main representatives that have been extensively studied with strong evidence that these molecules have potential as new therapeutic options for neural disorders [53-56].

New sentence: Endocannabinoid system (ECS) consists of endogenous cannabinoids the receptors and metabolic enzymes. This system is present in most vertebrate species and is distributed in several organs and tissues, including the CNS [54]. Endocannabinoids are endogenous bioactive lipid that are produced locally through specific biosynthetic pathways. Endocannabinoids are divided into two classes, N-acylethanolamines (NAEs) and primary amides. Anandamide and 2-arachidonylglycerol (2-AG) (Figure 2), are lipid mediators derived mainly from arachidonic acid, and are the most frequent biosynthetic routes [55-57]. Endocannabinoids, such as anandamide and its analogs, have been extensively studied as targets of new therapeutic options for disorders of the CNS, including the prevention of epileptic seizures [9,58]. 

  1. Page 4, line 180-181, clarify why the enzymes are highlighted. Also explain in details about these enzymes.

Reply: We thank the reviewer suggestion and added this discussion, as follows:

DAGL and phospholipase Cβ enzymes participates in the synthesis of 2-AG from membrane phospholipids. 2-AG is degraded mainly by MAGL enzyme, producing arachidonic acid (AA) and glycerol. FAAH degrades anandamide to AA and ethanolamine [56]. These enzymes have crucial roles in the regulation of the tissue levels or actions of endocannabinoids.

  1. Page 4, section 4, there was a discussion about the "Anandamide" however there is nothing discussed about 2-AG, which is focused in the section 6. I suggest the authors to discuss about 2-AG in detail next to the 3rd paragraph (after the discussion about anandamide)

Reply: We thank the reviewer suggestion and added this discussion, as follows:

2-AG is a monoacylglycerol molecule that acts as a retrograde messenger by activating both CB1 and CB2 cannabinoid receptors and is a key regulator of neurotransmitter release [56].

  1. page 4, section 4, there was a discussion on CB1, however, nothing was discussed about CB2, elaborate on it further.

Reply: We thank the reviewer suggestion and added this discussion, as follows:

CB2 receptors are seven transmembrane G protein-coupled receptors that are predominantly expressed in cells and tissues of the immune system [77]. However, CB2 has also been identified in areas of the CNS, mainly brainstem neurons, astrocytes and microglial cells [78]. The CB2 receptor is overexpressed in response to diverse CNS insult and may play an important role in epilepsy [79].

  1. Section 5 has to be rewritten focusing more on the endocannabinoids from plant oils. Most of the discussion is off topic and irrelevant. line 215 to 220 is more related to types of endocannabinoids, move this to section 4 at appropriate position.

Reply: We thank the reviewer suggestion and added this discussion, as follows:

The fatty acids of Patawa oil (Oenocarpus bataua), has also been submitted to the amidation process, with the major products being the endocannabinoid analogues N-isopropylpalmitamide and N-isopropyloleamide [93]. Oils extracted from Bertholletia excelsa (Brazil nut) have also been a substrate for the production of endocannabinoid analogues using lipase from Pseudomonas fluorescens, with a yield of up to 95 % after reaction [96].

  1. section 5, line 221 to 226, there was discussion focusing on synthesis of synthetic endocannabinoids, which is inappropriate in the context.

Reply: We thank the reviewer suggestion and the sentence was removed.

  1. Also, line 230 to 232, there was discussion about the authors work, which is not relevant to the discussion (better have a separate section about the synthetic cannabinoids and add more information, not just authors own work.

Reply: We thank the reviewer suggestion and the sentence was removed.

  1. Figures 1 and 2 use chemdraw to draw the chemical structures (not the pictures).

Reply: Done.

Round 2

Reviewer 2 Report

Authors did a good job in editing the manuscript except one comment which is copied below.

  1. Page 2, line 63 to 76, there was a discussion only on generalized seizures and their subtypes. However, there was no discussion given about the other two types of seizures (focal and epileptic spasms)

Please address this comment.

Author Response

We changed the paragraph and added a discussion about the other two types of seizures (focal and epileptic spams) as suggested.

According to the International League Against Epilepsy (ILAE), seizures are divided into three categories: generalized, focal (previously called partial), and epileptic spasms [12,13].

Generalized seizures start in neuronal networks in both hemispheres and can be subdivided into absence seizures and generalized tonic-clonic (GTC), myoclonic and atonic seizures. Absence seizures are characterized by sudden lapses of consciousness, blinking eyes or head movement, and their pathophysiology seems to be associated with rhythmic oscillations of the thalamus-cortical pathways with generalized spike and slow wave discharges and are generally associated with mutations of the Ca++ channel [14,15]. GTC seizures consist of bilateral symmetrical convulsive movements of all limbs in addition to impairment of consciousness, unlike myoclonic seizures, which consist of sudden and involuntary movements that can affect one or several muscles, just as atonic seizures involve weakness of the muscle tone followed by drop attack [16]. Myoclonic and atonic seizures initially show a normal electroencephalogram followed by generalized polyspike-and-wave epileptiform activity that precedes the onset of myoclonic-atonic and atonic seizures. The pathophysiology of these epileptic syndromes is more associated with voltage-gated Ca++ channels and the GABAA receptor α1 subunit [15,17]. 

Focal seizures originating in neuronal networks are limited to a single cerebral hemisphere and can develop at any point in life. Focal epilepsies include unifocal and multifocal disorders, as well as crises involving a variety of types of epileptic seizures, and may include focal perceptual seizures, focal perceptive or impaired perception, focal motor seizures, and non-motor and focal seizures progressing sometimes to bilateral tonic-clonic seizures [18,19]. The Interictal EEG typically shows focal epileptiform discharges and symptoms occur only when the seizure spreads to activate or disrupt cortical networks; as such, there is a wide spectrum of focal disease that depends not only on the location of the epileptic focus and its duration, but also on the connected cortical and subcortical areas [18,19].

Epileptic spasms are a type of epileptic syndrome in children with infantile spasms and shows change in cerebral rhythms defined as hypsarrhythmia, with high-amplitude, arrhythmic, asynchronous delta activities and multiple spikes. These epileptic spasms are associated with alterations in the gene regulation network of the GABAergic forebrain during development, and abnormalities in molecules expressed at the synapse [3,20,21].